The mountain papaya may be a possible reservoir of the Kashmir bee virus

Faúndez-Acuña Jorge Y. 1 2
Verdugo Diego 1 2
Vergara David 2
Olivares Gerardo 3
Ballesteros Gabriel I. 4 5
Quiroz Karla 2 3
Villarroel Carlos A. 2 3
González Gloria ggonzalez@ucm.cl 2 3
1 Doctorate in Translational Biotechnology (DBT), Catholic University of Maule , Talca , Maule Region , Chile
2 Center for Biotechnology of Natural Resources (CENBio), Catholic University of Maule , Talca , Maule Region , Chile
3 School of Biotechnology Engineering, Catholic University of Maule , Talca , Maule Region , Chile
4 Center of Integrative Ecology, University of Talca , Talca , Maule Region , Chile
5 Institute of Interdisciplinary Research, University of Talca , Talca , Maule Region , Chile
Roper James
Electronic publication date: 2025 Feb 21
Publication date: 2025
Volume: 13
Electronic Location ID: e18634
Received 2024 Aug 7; Accepted 2024 Nov 12
Copyright: ©2025 Faúndez-Acuña et al.
Copyright year: 2025
Copyright holder: Faúndez-Acuña et al.
License: This is an open access article distributed under the terms of the Creative Commons Attribution License, which permits unrestricted use, distribution, reproduction and adaptation in any medium and for any purpose provided that it is properly attributed. For attribution, the original author(s), title, publication source (PeerJ) and either DOI or URL of the article must be cited.
License URL: https://creativecommons.org/licenses/by/4.0/

Keywords: Chilean papaya, Kashmir bee virus, Sequencing, Transcriptome, Vasconcellea pubescens, Viral identification

Funding: FIC project “Fortalecimiento de la Competitividad en la Industria Papayera de la Región del Maule Mediante el Desarrollo de Herramientas Biotecnológicas” BIP: 40.001.007 The data used in this work was acquired with the financial support of the FIC project “Fortalecimiento de la Competitividad en la Industria Papayera de la Región del Maule Mediante el Desarrollo de Herramientas Biotecnológicas” BIP: 40.001.007. The funders had no role in study design, data collection and analysis, decision to publish, or preparation of the manuscript.

==============================
Background

The Kashmir bee virus (KBV) infects many species of Hymenoptera, including bees, wasps, and other pollinators, potentially contributing to honeybee population declines. KBV can cause death of bees. KBV can infect through both vertical transmission (from queen to larvae and vice versa) and horizontal transmission (via food contamination). Plants pollinated by bees may be a source of horizontal transmission, through fecal contamination of pollen and flowers by infected pollinators, both intra- and interspecifically. Pollinated plants constitute a source of KBV intra- and inter-species horizontal transmission, particularly by the contamination of pollen and flowers by feces of KBV-infected pollinators.

Result

We test for the presence of KBV sequences in the transcriptomes of Vasconcellea pubescens, a commercially valuable plant species known as mountain papaya. We mapped transcriptomes from fruit, leaves, and root tissues to the KBV reference genome with 91% coverage, from which we produced a consensus sequence denominated Kashmir bee virus ch. phylogenetic analysis revealed that KBV-Ch shares 97% nucleotide identity with the reference genome, and groups with other KBV strains isolated from Spain, Chile and New Zealand.

Introduction

Kashmir bee virus (KBV) is a positive-sense, single-stranded RNA virus classified in the Dicistroviridae family within the Cripavirus genus (Mazzei et al., 2019). KBV primarily infects bee species such as Apis cerana, Apis mellifera, and bumblebees (Bombus spp) within the Apidae family. It has also been identified in wasps (Vespula germanica) from the Vespidae family (De Miranda et al., 2004). Notably, KBV can co-infect with closely related viruses, such as acute bee paralysis virus (ABPV) and Israel acute paralysis virus (IAPV), forming the AKI viral complex (De Miranda et al., 2004; Evans & Schwarz, 2011). KBV isolates have been reported globally, including Spain, New Zealand, South Korea, North America, Australia, and Chile (Berényi et al., 2006; Nanetti et al., 2021; Riveros et al., 2018; Tentcheva et al., 2004).

The KBV reference genome spans 9,524 base pairs and contains two open reading frames (ORFs) that encode nonstructural and structural proteins (De Miranda et al., 2004). The 5′ ORF encodes the non-structural polyprotein, comprising three domains corresponding to a helicase, a 3C protease domain, and eight RNA polymerase domains that includes an RNA-dependent RNA polymerase. The 3′ ORF encodes a structural polyprotein composed of two capsid protein domains belonging to the VP4 superfamily (De Miranda et al., 2004).

In honeybees, KBV primarily infects worker bees, but can spread throughout the colony under stressful conditions. Transmission occurs vertically from queen to offspring, and horizontally among bees through contaminated food, affecting individuals from the larval stage to adulthood (Meeus et al., 2014). Additionally, Varroa destructor, a common mite pest of bees, can be a vector and activator of KBV and other viruses (Brødsgaard et al., 2000; Shen et al., 2005). Survival of pathogenic KBV particles in mites can explain how different mite species are potential routes of KBV (Carreck, Ball & Martin, 2010; Tixier, 2018). Moreover, increased stress levels in mite-infected hives makes KBV transmission more likely (De Miranda, Cordoni & Budge, 2010). Although primarily infecting bees within the Apidae family, KBV has been found in other Hymenopterans, including the German wasp Vespula germanica and the Asian hornet V. velutina, both of which can prey on A. mellifera (Brenton-Rule et al., 2018; Felden et al., 2020; Mazzei et al., 2019). The transmission of KBV to V. germanica is thought to be horizontal by feeding on infected live or dead honeybees (Eroglu, 2023; Evison et al., 2012).

Plants serve as potential reservoirs of entomopathogenic viruses that affect pollinators through contamination of pollen and flowers. Recently the pollen virome has been explored (Fetters et al., 2022), revealing a wide diversity of plant and insect pathogenic viruses. The shared use of flowers by different pollinator species may contribute to horizontal transmission routes (Durrer & Schmid-Hempel, 1994), often through feces deposition (Figueroa et al., 2019). Recent research has emphasized the role of flowers (Alger, Burnham & Brody, 2019), nectar, and pollen in facilitating inter-species viral transmission among pollinators, suggesting that plants may act as reservoirs for entomopathogenic viruses (Mcart et al., 2014). KBV has been detected in honeybee feces (Hung, 2000), parasitic mites (Shen et al., 2005), and pollen (Singh et al., 2010). Similarly, the closely related ABPV and IAPV have been found in flowers. Transmission of viruses between managed bees, wild bees, and other pollinating insects is common (Gisder & Genersch, 2017; Mcmahon et al., 2015). In addition, plants serve as vector of entomopathogenic viruses that affect insect herbivores, such as aphid-infecting RNA viruses (Jones, 2018).

The mountain papaya (Vasconcellea pubescens), also known as Chamburo, Chilean papaya, or wild papaya (Salvatierra-González & Jana-Ayala, 2016) belongs to the Caricaceae family, V. pubescens is a perennial, herbaceous, and trioecious species capable of cross-pollination, occasionally exhibiting hermaphroditism (Chong-Pérez et al., 2018). The fruit is smaller in comparison to other papaya species (Briones-Labarca et al., 2015).

Well-described phytopathogenic viruses infecting Carica papaya include papaya ringspot virus, papaya leaf distortion mosaic virus, papaya lethal yellowing virus, and papaya mosaic virus, among others (Abreu et al., 2015; Adams, Antoniw & Beaudoin, 2005; Chávez-Calvillo et al., 2016; Razean Haireen & Drew, 2014; Yang et al., 2012). However, the virome of V. pubescens remains largely unexplored. Documenting viruses affecting this species is crucial due to their potential impact. Additionally, although not directly affecting papaya, the presence of entomopathogenic viruses could impact pollinating insects, such as bees, which interact with papayas (Badillo-Montaño, Aguirre & Munguía-Rosas, 2019). Therefore, establishing a viral library of mountain papaya could benefit both the agroindustrial and apiculture sectors.

In this study, we conducted transcriptome analyses of leaf, root, and fruit tissues to establish a virome library of the mountain papaya collected from fields in central Chile. This is the first report identifying the presence of KBV viral sequences in mountain papaya. The detection of KBV viral sequences in roots, leaves, and in vitro cultivated plants suggests that V. pubescens may often be a reservoir or vector for KBV infection.

Materials & Methods

Material collection

Plant samples were collected from V. pubescens orchards in Lipimávida (34°51′4.7″S, 72°8′27.6″W), Licantén (34°59′7.1″S, 71°59′4.9″W), Vichuquén (34°52′59.9″S, 71°59′36.2″W), Chanco (36°16′8.4″S, 72°42′57.2″W), Pelluhue (35°48′49.0″S, 72°34′26.4″W), Putú (35°12′47.9″S, 72°17′2.0″W), Iloca (34°55′57.7″S, 72°10′50.5″W), Duao (34°53′48.8″S, 72°10′46.6″W), Constitución (35°19′49.4″S, 72°24′33.1″W), and Curanipe (35°50′37.7″S, 72°38′21.1″W), located in the coastal areas of the Maule region, Chile (Fig. 1). In total, 100 orchards were sampled, with 15 plants selected from each orchard, and three samples taken per tissue type (leaves, roots, and fruits). Plant material was randomly selected using a zigzag pattern, covering the entire area of the orchard. Each tissue sample, including leaves, roots, and fruits, was collected in triplicate.

Figure 1 Map of the Maule Region depicting Vasconcellea pubescens cultivars.

Red stars, from top to bottom, indicate the cities of Lipimávida, Duao, Iloca, Putú, Constitución, Chanco, Pelluhue, and Curanipe. Green stars, from top to bottom, indicate the cities of Vichuquén and Licantén. The zoomed-in area indicates the samples obtained from each locality, corresponding to roots, leaves, and fruits. Coordinates of the North Extreme: 17.5083°S, 69.6116°W and Coordinates of the South Extreme: 56.5000°S, 68.1333°W. Map source credit: ©Contributers of OpenStreetMap, ODbL.

RNA extraction and sequencing

RNA extraction from papaya fruit, leaves, and roots was performed using the Spectrum™ Plant Total RNA Kit (Sigma-Aldrich, St. Louis, MO, USA). Quantification of concentration and purity was carried out using the NanoDrop 2000 equipment (Thermo Fisher Scientific, USA). Total RNAs were pooled in equal amounts (2 µg, RIN > 7–8) from 450 samples to generate a mixed cDNA library of V. pubescens. Nine cDNA libraries were sequenced in paired-end mode with an Illumina HiSeq™ 2000 sequencer. Transcriptomes from leaves, roots, and fruits were sequenced. These transcriptomes were processed using the TruSeq Stranded mRNA LT Kit (Illumina) at Genoma Mayor (Chile) and sequenced with Illumina technology. Sequencing data has been deposited in NCBI SRA under the PRJNA1142012 accession number.

Preliminary treatment of transcriptomes

FastQC was employed to assess the quality of raw sequences obtained with Illumina (v0.12.0) (http://www.bioinformatics.babraham.ac.uk/projects/fastqc/). Trimmomatic was utilized to remove adapter sequences and low-quality reads (reads with ambiguous bases ‘N’ and reads with more than Q < 20 bases) to obtain high-quality reads (v0.33) (Bolger, Lohse & Usadel, 2014; González, Aguilera & D’Afonseca, 2020). The Trinity software was used for de novo assembly from the concatenated high-quality reads of these libraries (v2014-04-13) (Grabherr et al., 2011), utilizing default parameters.

Transcriptome annotation

ORFinder (https://www.ncbi.nlm.nih.gov/orffinder/), blastn, and blastp (https://blast.ncbi.nlm.nih.gov/Blast.cgi) against nt/nr from NCBI were utilized to predict open reading frames (ORFs) from the assembled sequences (https://www.ncbi.nlm.nih.gov/orffinder/). ORFs were compared with reference genomes to identify the start (methionine) and stop codon. The following tools were employed for functional annotation: CDD (https://www.ncbi.nlm.nih.gov/Structure/cdd/wrpsb.cgi/) (Marchler-Bauer et al., 2015), PFAM (https://pfam.xfam.org/) (El-Gebali et al., 2019), SMART (http://smart.embl-heidelberg.de/) (Letunic & Bork, 2018), and PROSITE (https://prosite.expasy.org/) (Sigrist et al., 2013) for predicting conserved protein domains; PLACE (https://www.dna.affrc.go.jp/) (Higo et al., 1999) and Neural Network Promoter Prediction (https://www.fruitfly.org/seq_tools/promoter.html) (Reese, 2001) for predicting promoters and regulatory regions; and ELM (http://elm.eu.org/) (Dinkel et al., 2012; Elkhaligy et al., 2021) for profile identification.

Viral identification

Viruses were first identified in the transcriptome by exploring the taxonomy of the blast annotated sequences. Four putative viral contigs were compared with sequences from the NCBI database, which showed a match with Kashmir bee virus (AY275710.1).

Reference mapping and consensus sequence generation

Mapping to the KBV reference genome (AY275710.1) was conducted for the nine transcriptome samples using BWA with default parameters. Subsequently, a consensus sequence for the putative KBV was obtained using the bam alignment of one sample fruit tissue replicate three with the IVAR consensus tool (-n N -m 1 -t 0.5). The consensus sequence was named “Kashmir Bee Virus Chile isolate from Chile (KBV-Ch)” and it was submitted to NCBI with the accession PQ820963.1.

Phylogenetic analysis

A phylogenetic analysis was conducted to compare KBV-Ch sequence to other KBV Chilean isolates using a 320 bp region that comprises the 3′ end of the intergenic region and the 5′ end of the structural polyprotein. In addition, a BLASTN of this 320 bp region to the nt database was performed to identify other KBV sequences to include in the phylogenic analysis. A total of 30 sequences were aligned using Muscle (default parameters), after which a phylogenetic tree was constructed using the Neighbor-Joining method in MEGA (Kimura 2 parameter method, 1,000 bootstraps). The Israel Acute Paralysis Virus was included as an outgroup. The sequences used were EU122368.1 (USA), EU122369.1 (USA), EU122370.1 (USA), EU122371.1 (USA), EU122372.1 (USA), EU122373.1 (USA), NC_004807.1 (USA), MT096516.1 (Spain), KC833152.1 (Chile), KC833158.1 (Chile), KC833142.1 (Chile), KC833149.1 (Chile), AY661447 (New Zealand), MW314660.1 (Spain), PQ820963.1 (Chile), MN226368.1 (Nigeria), EU122374.1 (USA), MN226367.1 (USA), HM067845.1 (USA), KP965377.1 (South Korea), KP965379.1 (South Korea), KP965382.1 (South Korea), KP965383.1 (South Korea), KP965378.1 (South Korea), KP965376.1 (South Korea), KP965381.1 (South Korea), KP965380.1 (South Korea), KP965375.1 (South Korea), KP965374.1 (South Korea), KP965373.1 (South Korea), KF219804.1 (Israel).

Plant material and in vitro thermotherapy

Nodal explants were obtained from papaya plants in the Lipimavida area (34°51′4.7″S, 72°8′27.6″W), coastal Region of Maule, and established in vitro in culture tubes containing 10 mL of Murashige and Skoog (MS, 1962) medium at 100% concentration supplemented with 30 mg/L sucrose, 0.5 mg/L Benzylaminopurine (BAP), and 0.4 mg/L Indole-3-acetic acid (IAA), adjusted to a pH of 5.8. These were cultivated in a growth chamber at 24 °C ±  1 °C and a 16-hour photoperiod provided by 24-Watt white fluorescent tubes, generating a light intensity of 60 µmol/m2/s. A total of 60 plants were sanitized using thermotherapy carried out in a growth chamber (Faithfull, RGX-400EF) for a period of 4 weeks at a temperature of 38 °C ± 0.5 and a 16-hour light/8-hour dark photoperiod. After the heat treatment, shoot tips (five mm) were excised and transferred to culture flasks containing 30 ml of MS medium supplemented with 30 mg/L sucrose and 0.4 mg/L Zeatin, adjusted to a pH of 5.8. A total of 20 plants survived, from which 7 clones were generated, of these four were used for RNA extraction.

RNA extraction and RT-PCR analysis

Four line in vitro cultured of V. pubescens (T1-T4) were used for RT-PCR identification of KBV-Ch. 100 mg of each sample was processed with liquid nitrogen to extract RNA using the Spectrum TM Plant Total RNA Kit (Sigma-Aldrich) according to the manufacturer’s instructions, then cDNA was synthesized using the Thermo Scientific TM Revert Aid First Stand cDNA Synthesis Kit (Sigma-Aldrich). Subsequently, a solution of 12 µL of 1X PCR Buffer (Winkler, Chile), 1.5 mM PCR Buffer (Winkler, Chile), 2.5 mM MgCl2 (Invitrogen, Thermo Fisher Scientific), 0.2 mM dNTPs (Invitrogen, Thermo Fisher Scientific), along with 300pM of each primer and 1.5u Taq DNA polymerase (Agilent) was prepared. The MultiGene™ OptiMax thermocycler (Labnet™) was used and an initial denaturation program of 5 min at 95 °C, followed by 35 cycles of denaturation for 30 s at 95 °C, alignment for 30 s at 58 °C and a final extension of 30 s at 72 °C, remaining at 4 °C until verification by 2.3% m/v agarose gel electrophoresis in 1X TAE buffer (Winkler, Chile). GeneRuler™ Ready-to-use 50 bp DNA Ladder molecular weight marker (Thermo Fisher Scientific) and GeneRuler™ Ready-to-use 100 bp Plus DNA Ladder (Thermo Fisher Scientific). The primers for amplify the KBV sequence were forward 5′-ATGATTGGGGGGCGGTGTAATA-3′ and reverse 5′-TGCCTGTGTGAAAAGCTGTC-3′ to obtain a 209 bp amplicon that target a conserved region of the VP2 protein (8,023-8, 231 bp; KBV reference genome AY275710.1). In addition, primers for V. pubescens ribosomal 18s RNAr were used as positive control (Gambino & Gribaudo, 2007).

Results

Detection of Kashmir bee virus sequences in V. pubescens transcriptomes

De novo assemblies of root, leaf, and fruit transcriptomes from V. pubescens samples collected across multiple orchards in the Maule region (Chile) revealed diverse viral sequences, contributing to the characterization of its virome. Blastp annotation to the nr database revealed contigs that match 235 KBV. We then compared the viral load across different tissues by mapping transcriptome reads to the KBV reference genome (Table 1). Notably, one replicate of fruit samples showed 820 mapped reads, covering 92% of the reference sequence with an average depth of 12X. In addition, KBV was also detected in root and leaf samples ranging from two to six reads per sample (Table 1). From the reads aligned to the KBV reference genome, we constructed a consensus sequence named “KBV-Ch (Chile)”, with 199 single nucleotide polymorphisms (SNPs) (Table S1).

Table 1 KBV detection and RNA sequencing in mountain papaya tissues.

Summarizes the number of reads in Kashmir bee virus (KBV) genomes in different mountain papaya (V. pubescens) tissues. Samples of fruits, leaves and roots were analyzed in triplicate.

	Number of reads mapped to KBV genome	Total reads	
Root R1	6	46,122,074	
Root R2	2	47,440,504	
Root R3	2	39,754,722	
Leaf R1	1	55,401,252	
Leaf R2	0	42,124,523	
Leaf R3	0	72,734,557	
Fruit R1	0	51,194,927	
Fruit R2	0	45,857,234	
Fruit R3	820	43,542,557	

Phylogenetic analysis

KBV has been previously reported in honeybee samples in Chile (Riveros et al., 2018). Phylogenetic analysis which revealed that KBV-Ch is similar to a KBV isolate from Spain, and sharing a clade with isolates from New Zealand and some samples from Chile (Fig. 2).

Figure 2 Evolutionary relationships among the various KBV strains.

The phylogenetic tree of KBV Ch based on a 320 bp of the intergenic region and structural polyprotein gene from twenty-nine distinct sequences of KBV. The isolates originate from USA, North Korea, Chile, Spain, Nigeria, and New Zealand. The Israel Acute Paralysis Virus (KF219804) was used as an outgroup. This Chilean isolate, PQ820963.1 KBV Ch, are highlighted in red. The dendrogram was constructed using the Neighbor-Joining method in MEGA and 1,000 Bootstrap replicates.

Detection of KBV in laboratory and field plants

Given the detection of KBV sequences in root and leaf tissues, suggesting potential vascular tissue incorporation of the virus, we investigated the presence of KBV in in vitro cultivated plants derived from V. pubescens explants collected from the field using RT-PCR. Examination of in vitro cultivated plants found a single, 209 bp, amplicon in one sample (Fig. 3).

Figure 3 KBV sequences are detected in in vitro grown wild papaya.

RT-PCR amplification results of KBV Ch in V. pubescens explants. (Left) Housekeeping 18S rRNA gene from V. pubescens (∼844 bp) as a positive control in four independent samples (T1–T4), negative control (c), and GeneRuler™ 100 bp Plus DNA Ladder. (Right) KBV Ch sequence (∼209 bp) in four independent samples (T1–T4), negative control (c), and GeneRuler™ 50 bp DNA Ladder. Images were obtained using Accuris E3000 UV Transilluminator.

Discussion

KBV was identified in the mountain papaya through transcriptome analysis of several tissues, and KBV sequences were found in greenhouse-grown plants. Through transcriptome analysis of leaf, root, and fruit tissues of the mountain papaya, V. pubescens, we successfully identified a near-complete genome sequence of the KBV, and detected KBV sequences in in vitro grown plants. The presence of KBV was notably higher in one replicate of fruit transcriptome, suggesting an occasional ocurrence of the virus in V. pubescens orchards. As a trioecious species capable of cross-pollination, V. pubescens produces male, female, and hermaphrodite plants (Carrasco et al., 2009). Pollination can occur via wind or insects, with self-pollination observed in commercial varieties of Carica papaya (Carrasco et al., 2022). Infected pollen presents a potential source of KBV particles in fruits. Some viruses can infect pollen, potentially transferring the infection to the plant ovule and resulting in virus-infected fruit. This infection in pollen can occur through various vectors, including abiotic factors (such as wind) and biotic factors (such as insects), suggesting a possible mechanism for virus transmission from pollen to fruit (Bhat & Rao, 2020). In addition, Figueroa et al. (2019) studied the mechanisms involved in pathogen deposition, persistence and acquisition in flowers. Bees, known to spend extended periods feeding on flowers, deposit and acquire feces from other pollinators, promoting transmission rates.

While reports on bee viruses in pollinated plants typically focus on flowers and pollen, other viruses from the Dicistroviridae family have been detected circulating in other plant tissues, among which most examples are from aphid viruses (Jones, 2018). For instance, the Rhopalosiphum padi virus (RhPV) spreads systemically in barley plants including roots, potentially infecting healthy individuals that consume RhPV-carrying plants (Ban et al., 2007). Similarly, the aphid lethal paralysis virus has been found in cucumber leaf transcriptomes, suggesting a comparable transmission route (Maina et al., 2017). We detected KBV RNA in roots and leaves, albeit in lower quantities compared to fruits, suggesting that KBV could circulate from contaminated sexual organs to other plant tissues. Despite rigorous surface cleaning of plant material before RNA extraction, residual KBV RNA could stem from superficial contamination with feces, corpses, or other insect sources carrying KBV particles. Importantly, KBV RNA was also detected in in vitro grown plants, which underwent stringent disinfection procedures, and only novel tissue was used for RNA extraction. Virus in in vitro plant cultures can originate from the source vegetative plant (Li et al., 2013). Such is the case of babaco plants (Vasconcellea x heilbornii), which are often propagated from cuttings without virus elimination protocols typically resulting in in vitro plants infected with viruses (Muñoz et al., 2023). Therefore, it is possible that KBV could systemically circulate through the plant as it was shown for other members of Dicistroviridae, utilizing pollinated plants as reservoirs and passive vectors. The phenomenon of horizontal transmission of entomopathogenic viruses by host plants as passive vectors remains understudied yet holds significant implications for pest control and conservation of wild pollinators (Jones, 2018).

While the exact mechanism of KBV introduction into V. pubescens populations in Chile remains uncertain, previous studies conducted by Chilean researchers have identified partial KBV sequences Apis mellifera populations across several regions in Chile (Riveros et al., 2018). Phylogenetic analysis of the KBV sequence obtained in our study reveals close genetic proximity to these previously reported Chilean isolates, as well as isolates from Spain and New Zealand, suggesting reintroduction of genetically different KBV isolates across continents. Furthermore, the interaction between bees and papaya trees is noteworthy (Badillo-Montaño, Aguirre & Munguía-Rosas, 2019). This interaction implies two key aspects: firstly, bees in the studied locations may act as vectors for KBV transmission, potentially disseminating the virus during pollination of V. pubescens; and secondly, there is a critical need to monitor bee populations in the coastal areas of the Maule Region for KBV presence. Such surveillance could benefit beekeepers and lay the groundwork for forthcoming investigations.

Conclusions

This study marks a significant advancement by documenting, for the first time, the detection of the “Kashmir Bee Virus Chile” within the transcriptomes of Vasconcellea pubescens, representing a milestone for the agro-industrial and apiculture sectors in Chile’s Maule region. Transcriptome analysis revealed the presence of KBV-Ch in various V. pubescens tissues, with a notable abundance in fruit tissues, suggesting a potential role as a reservoir or vector of infection. These preliminary results highlight the presence of KBV-Ch in V. pubescens, serving as a fundamental step for the development and validation of future hypotheses. Additional replication and testing in more plants are necessary. Future studies should focus on elucidating transmission mechanisms and assessing the virus’s impact on agriculture and beekeeping. These efforts are critical for developing strategies to control viral diseases affecting crops and pollinators.

Supplemental Information

Supplemental Information 1 SNPs found in KBV-Ch

The single nucleotide polymorphisms (SNPs) identified in the KBV-Ch (Chile) consensus sequence and the position of each SNP in the reference genome, the reference allele, and the corresponding KBV-Ch allele.

We would like to express our gratitude to the entire team of the papaya project at the Center for Biotechnology of Natural Resources (CENBio) at the Universidad Católica del Maule.

Additional Information and Declarations

Competing Interests

Author Contributions

Data Availability

The authors declare there are no competing interests.

Jorge Y. Faúndez-Acuña performed the experiments, analyzed the data, prepared figures and/or tables, authored or reviewed drafts of the article, and approved the final draft.

Diego Verdugo performed the experiments, authored or reviewed drafts of the article, and approved the final draft.

David Vergara performed the experiments, authored or reviewed drafts of the article, and approved the final draft.

Gerardo Olivares performed the experiments, authored or reviewed drafts of the article, and approved the final draft.

Gabriel I. Ballesteros analyzed the data, authored or reviewed drafts of the article, and approved the final draft.

Karla Quiroz performed the experiments, authored or reviewed drafts of the article, and approved the final draft.

Carlos A. Villarroel analyzed the data, prepared figures and/or tables, authored or reviewed drafts of the article, and approved the final draft.

Gloria González conceived and designed the experiments, authored or reviewed drafts of the article, and approved the final draft.

The following information was supplied regarding data availability:

The data is available at NCBI: PRJNA1142012 and PQ820963.1.

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
