# Peer review of "The mountain papaya may be a possible reservoir of the Kashmir bee virus"

_PeerJ, doi:10.7717/peerj.18634_

## Round 0.1 · original submission · Major Revisions

Your study appears to be an interesting and surprising examination and findings of KBV with a plant as a vector. However, there are several issues that one of the reviewers brought up that should be clarified. Also, I make a few notes in my annotated manuscript that you can read. I pointed out a few points where you could improve the English, some of which are also simply about better writing (not just in English). I think that following the reviewer's and my comments and suggestions your manuscript could be easily improved.

Reviewer 2 offered some very constructive suggestions, with which I agree. Rather than repeat those comments here, just be sure to read those suggestions closely and in your rebuttal explain clearly.

I hope you find these comments useful in your revisions.

Reviewer 1 ·

Basic reporting

Your article generally complies with the professional article structure; the text content of the figures and tables needs to be improved and more detailed. Revize their explanation corresponding to the text and add marks on electrophoresis gel figures.
The material methods section needs detail. Please explain the sampling details (lines 104-111), RNA extraction (lines 114-117), invitro cultivation conditions of plants (line: 174), electrophoresis and RT-PCR (lines: 184-188)

Experimental design

No comment.

Validity of the findings

Conclusions are well stated and examined, linked to the original research question.
Your research question was well-defined. The question "Are the mountain papaya of Chile reservoir for KBV?" has been interrogated in your study. While your results are compelling, the supporting evidence is limited. To bolster your hypothesis, it's crucial to provide more proof that you have eliminated surficial virus contamination from the mountain papaya plant samples. The lack of a detailed plant culture methodology for in vitro production leaves your results open to criticism, which could undermine their reliability and the reproducibility of your work. Additionally, the choice not to use the negative sense viral genome strand-specific PCR method for definitive proof of the virus's presence in the plant samples needs to be justified.

Reviewer 2 ·

Basic reporting

No comment

Experimental design

Faúndez-Acuña and collaborators in "The mountain papaya: Is it a possible reservoir of the Kashmir bee virus?" present interesting findings regarding the presence of the insect virus Kashmir bee virus (KBV) in the fruit tissue of mountain papaya samples. The manuscript has several strengths, including a compelling research question and a thorough exploration of transkingdom novel host-virus interaction.
However, to strengthen the manuscript, the following recommendations should be addressed:

The most crucial issue is the experimental design and result replicability. The most critical issue lies in the experimental design, particularly the replicability of the results. The manuscript mentions that only one fruit replicate KBV was detected through RNAseq analysis, which raises concerns about technical issues during sequencing or might indicate potential contamination. Providing evidence as to why these results are robust is necessary to withhold the hypothesis. Additionally, more clarity is needed regarding the tissue sampling. Were the sample replicates for RNAseq pooled or individually sequenced? If pooled, that might explain why only one sample resulted positive for KBV.
I attempted to calculate the sequencing depth for the RNAseq experiment, but the methodology details, such as sequencing length and single vs paired-end method, needed to be included. This information is critical to assess the reliability of the sequencing data, especially given that only a small number of reads (six, two, and two from roots, one from leaves) were mapped to KVB. Given the low read count, I recommend caution when labeling these tissues as positive for KBV to claim that additional evidence is needed (for example, RT-PCR). The RT-PCR analysis included only four samples from in vitro tissue culture, which is too few to draw robust conclusions. Are any plants from the ten sample locations included in this assay? Expanding the RT-PCR testing to the ten locations and RNA used for the RNAseq would strengthen the conclusions.
Furthermore, while it is promising that some evidence of KBV was found in V. pubescens, the current data are not conclusive enough to assert that this species is a KBV reservoir. Additional replicates or testing of more plants would be needed.

The sequence and bioinformatic analysis descriptions in the materials and methods need better detail, and some methodological aspects must be clarified. For instance, it is mentioned that viral contigs were identified. However, it needs to be clarified whether all the contigs were subjected to BLAST searches or only the four putative viral ones. What databases were used for the BLAST searches?
Additionally, the way in which the virus genome KBV-Ch was generated needs to be clarified. Was it generated from the different RNAseqs or only the fruit RNAseq with the most reads? This would provide a clearer picture of how robust the genome assembly is. Also, the authors mention the viral genome organization in the introduction, but this could be presented as a result, with explicit annotation of the genes, and indicate the 199 polymorphisms (if possible) in KBV-Ch. Also, the primer positions used in this study should be indicated, as it would help explain the genomic region used for amplification by RT-PCR.

Validity of the findings

No comment, see other sections.

Additional comments

The introduction does an adequate job setting up the role of pollen in spreading viruses to pollinators, but it could benefit from tighter integration of key concepts. Specifically, there needs to be a stronger connection between the virus ecology and epidemiology, insect host range, and global distribution. For example, provide context by expanding on how KBV primarily infects bees within the Apidae family, such as Apis cerana, A. mellifera, and bumblebees (Bombus spp.), but has also been found in wasps (Vespula germanica, Vespidae, de Miranda et al., 2004). Transmission of KBV
occurs both vertically from queen to offspring and horizontally among bees through contaminated food or vectors like the common mite Varroa destructor, affecting individuals from larval stages to adulthood (Brødsgaard et al., 2000; Shen et al., 2005; Meeus et al., 2014). Highlighting these transmission dynamics can better frame the study and focus on KBV's importance and spread in the colony.

Moreover, while the authors mention the role of pollen, it would be useful to cite other examples where insect viruses have been found in pollinators and pollen (Singh et al., 2010) or insect viruses replicating in plants (Kormelink et al., 2011; Jiwaji et al., 2019). This provides a broader epidemiological context. Also, it would be insightful to include a more detailed description of pollinators associated with Vasconcellea pubescent and their role in pollination, which could support the hypothesis on virus transmission to pollinators.

Finally, the purpose and objectives of the study should be explicitly stated at the end of the introduction. For example, the rationale for why KBV might be present in mountain papaya as a reservoir could be more clearly linked to the hypothesis and research goals. This will also emphasize the significance of the findings.


Several minor issues should be addressed for consistency and clarity.
• For naming convention, consider changing the name to "Kashmir bee virus isolate from Chile (KMV-Ch)," a more standard nomenclature in virology.
• Materials and methods: Details regarding the kits and reagents are missing. Please include company specifications. Remove the acknowledgments in the material collection subsection.
• In Table 1, the 0% column could either be removed or adjusted to include more decimals for clarity.
• In Figure 3, ensure that each sample's lanes in the gel electrophoresis image are clearly labeled.


References:

Jiwaji M, Matcher GF, de Bruyn M-M, Awando JA, Moodley H, Waterworth D, et al. (2019). Providence virus: An animal virus that replicates in plants or a plant virus that infects and replicates in animal cells? PLoS ONE 14(6): e0217494. https://doi.org/10.1371/journal.pone.0217494

Manley, R., Boots, M. and Wilfert, L. (2015). REVIEW: Emerging viral disease risk to pollinating insects: ecological, evolutionary and anthropogenic factors. J Appl Ecol, 52: 331-340. https://doi.org/10.1111/1365-2664.12385

---

## Round 0.2 · Minor Revisions

In this version, you have done well improving the quality of the text and addressing the issues brought up by the reviewers and myself. I am including the annotated manuscript with many suggestions to help you improve the quality of the English. My comments are not comprehensive, so please try to use these examples to extrapolate to other places in your text that might be improved.

Also, double-check your manuscript for formatting requirements. In this version, you did not indent your paragraphs, for example, which made some reading more difficult.

---

## Round 0.3 · accepted · Accept

There are a few minor details that you should still adjust. Your title, with the colon, is poorly written as I noted in my previous review. Please consider fixing that. In your legend of figure 1, I also suggested that you used the word "indicate" instead of "represent." I recommend, again, that you change that as well.